# Mistletoe-Extract Drugs Stimulate Anti-Cancer Vγ9Vδ2 T Cells

**DOI:** 10.3390/cells9061560

**Published:** 2020-06-26

**Authors:** Ling Ma, Swati Phalke, Caroline Stévigny, Florence Souard, David Vermijlen

**Affiliations:** 1Department of Pharmacotherapy and Pharmaceutics, Université Libre de Bruxelles (ULB), 1050 Bruxelles, Belgium; ling.ma@ulb.ac.be (L.M.); swatiphalke08@gmail.com (S.P.); Florence.Souard@ulb.be (F.S.); 2Institute for Medical Immunology, Université Libre de Bruxelles (ULB), 6041 Gosselies, Belgium; 3RD3 Department-Unit of Pharmacognosy, Bioanalysis and Drug Discovery, Université Libre de Bruxelles (ULB), 1050 Bruxelles, Belgium; Caroline.Stevigny@ulb.be; 4DPM UMR 5063, CNRS, Université Grenoble Alpes, 38041 Grenoble, France

**Keywords:** gammadelta, Vgamma9Vdelta2, herbal drug, mistletoe, T cell receptor, TCR

## Abstract

Human phosphoantigen-reactive Vγ9Vδ2 T cells possess several characteristics, including MHC-independent recognition of tumor cells and potent killing potential, that make them attractive candidates for cancer immunotherapeutic approaches. Injectable preparations from the hemi-parasite plant *Viscum album* L. (European mistletoe) are commonly prescribed as complementary cancer therapy in European countries such as Germany, but their mechanism of action remains poorly understood. Here, we investigated in-depth the in vitro response of human T cells towards mistletoe-extract drugs by analyzing their functional and T-cell-receptor (TCR) response using flow cytometry and high-throughput sequencing respectively. Non-fermented mistletoe-extract drugs (AbnobaViscum), but not their fermented counterparts (Iscador), induced specific expansion of Vγ9Vδ2 T cells among T cells. Furthermore, AbnobaViscum rapidly induced the release of cytotoxic granules and the production of the cytokines IFNγ and TNFα in Vγ9Vδ2 T cells. This stimulation of anti-cancer Vγ9Vδ2 T cells was mediated by the butyrophilin BTN3A, did not depend on the accumulation of endogenous phosphoantigens and involved the same Vγ9Vδ2 TCR repertoire as those of phosphoantigen-reactive Vγ9Vδ2 T cells. These insights highlight Vγ9Vδ2 T cells as a potential target for mistletoe-extract drugs and their role in cancer patients receiving these herbal drugs needs to be investigated.

## 1. Introduction

Recently, T cell-based cancer immunotherapy has become a main therapy arm in the clinic besides surgery, radio- and chemotherapy. While mostly conventional αβ T cells are considered, it has become increasingly clear that γδ T cells have a large potential, which is illustrated by the interest of commercial partners [1]. The γδ T cells express a T cell receptor (TCR) on their cell surface that is composed of a γ and a δ chain and the TCR-dependent recognition of cancer cells usually do not rely on classical MHC molecules. The antitumor function of γδ T cells is generally associated with their cytotoxic potential and their production of both interferon γ (IFNγ) and tumor necrosis factor α (TNFα) [1,2]. The γδ TCR is generated by the somatic recombination of the TRG and TRD loci, combining different variable (V), diversity (D, only in TRD), and joining (J) gene segments [3,4]. This combinational diversity together with junctional diversity can result in a large variation in possible V(D)J junctions, also called complementary-determining-region-3 (CDR3), the part of the TCR that usually plays an important role in antigen recognition. γδ T cells can be divided into subsets based on the type of V gene segment used in their γδ TCR. A major γδ T cell population in adult human peripheral blood is represented by Vγ9Vδ2 T cells that are defined by the expression of a TCR containing the γ-chain variable region 9 (Vγ9, TRGV9) and the δ-chain V region 2 (Vδ2, TRDV2). They are poised towards a cytotoxic type 1 effector phenotype: they can kill target cells through the degranulation of cytotoxic granules (containing perforin and granzymes) and they produce swiftly IFNγ and TNFα upon activation [1]. Vγ9Vδ2 T cells respond (expansion, release of cytotoxic granules, cytokine production) in a TCR-dependent manner towards small metabolites derived from the isoprenoid pathway, such as the endogenous-derived isopentenyl pyrophosphate (IPP) or the microbial-derived (E)-4-Hydroxy-3-methyl-but-2-enyl pyrophosphate (HMBPP), that critically depends on the transmembrane protein butyrophilin3A1 (BTN3A1) [5,6]. The Vγ9Vδ2 T cell subset has been a target for cancer clinical trials, mainly through their in vivo activation via the administration of aminobisphosphonates such as zoledronate. Aminobisphosphonates inhibit the farnesyl pyrophosphate synthase enzyme (FPPS) leading to the intracellular accumulation of endogenous IPP [1,7]. Interestingly, Vγ9Vδ2 T cells also respond to alkylamines such as *sec*-butylamine that are found in edible plants and tea [8,9]. These alkylamines, like aminobisphosphonates, activate Vγ9Vδ2 T cells indirectly through inhibition of the FPPS enzyme [10]. The stimulation of the antitumor function of γδ T cells by such plant-derived compounds are thought to play an important role in the prevention of cancer development [11,12,13].

Injectable preparations from the hemi-parasite plant *Viscum album* L. (European mistletoe) are the most frequently prescribed complementary cancer therapy in central European countries. For example, in Germany, up to 77% of cancer patients apply this therapy in the context of integrative oncological approaches and thus belong to the most prescribed drugs offered to cancer patients [14,15,16]. Mistletoe extracts are prepared as aqueous solutions and they can either be fermented (e.g., Iscador) or unfermented (e.g., AbnobaViscum) [15,17]. In addition, the commercial products can be subdivided according to the species of host tree, which is typically indicated in the product name by a suffix letter, such as P for pini (pine) and M for mali (apple tree). The different types of preparations and different host trees may contribute to variable contents of biological active substances [15]. In vitro studies and animal tumor models have shown that mistletoe extracts can be cytotoxic and immunomodulatory, but their precise mode of action is poorly understood [15,17,18,19].

Mistletoe treatment has been suggested to increase the survival of cancer patients, but this is controversial and thus an increased understanding of its mechanism of action is needed to guide further in vivo studies and clinical trials [14,15,19]. Here we show that non-fermented mistletoe-extract drugs (AbnobaViscum) stimulate and expand specifically Vγ9Vδ2 T cells, induce the release of cytotoxic granules and promote the production of the cytokines IFNγ and TNFα. Furthermore, we show that this mistletoe-mediated activation of anti-cancer Vγ9Vδ2 T cells is rapid and direct (i.e., not dependent on the accumulation of endogenous phosphoantigens) and is completely BTN3A-dependent.

## 2. Materials and Methods

### 2.1. Sample Collection

Peripheral blood mononuclear cells (PBMC) from healthy adult donors (> 18 years) were isolated from blood donations at the CHU Tivoli (La Louvière, Belgium) and included informed consent of the donors (Ethics Commission CHU Tivoli, ethical code number 917, 29 October 2013). The study was conducted in accordance with the Declaration of Helsinki. PBMC were isolated by Lymphoprep gradient centrifugation (AxisShield, Dundee, UK) and cryopreserved in liquid nitrogen.

### 2.2. Cell Culture and Treatments

PBMC were thawed at 37 °C in complete medium [(RPMI 1640 (Gibco, Invitrogen, Waltham, MA, USA), supplemented with L-glutamine (2 mM), penicillin (50 U/mL), streptomycin (50 U/mL), and 1% nonessential amino acids (Lonza, Basel, Switzerland) and 10% (*v*/*v*) heat-inactivated FCS (PPA Laboratories, Toronto, ON, Canada)] and cultured in 96-well round bottom plates at 2 × 10^5^ cells/well (1 × 10^6^ cells/mL). For activation tests (CD69 and proliferation), PBMC were rested for 2 h or overnight after thawing, then the stimulating compounds were added at following concentrations: HMBPP (10 nM, Echelon Bioscience, Salt Lake City, UT, USA), zoledronate (1 μM, Novartis, Basel, Switzerland), sec-butylamine (0.5 mM, Sigma-Aldrich, St. Louis, MO, USA), mistletoe extracts (1000 μg/mL). AbnobaViscum Pini and AbnobaViscum Mali were obtained from Abnoba GmbH (Pforzheim, Germany) and Iscador Pinus and Iscador Malus from Iscador AG (Arlesheim, Switzerland). The stock concentration was for all mistletoe-extract drugs 20 mg/mL. Cells were cultured at 37 °C and 5% CO_2_. For the CD69 readout, the cells were cultured for 1 day and tested. For the proliferation assay, PBMC were either cultured in the presence of heat-treated mistletoe extracts for 7 days or washed 3 times after 1 day of incubation with non-heated extracts to remove stimulants (pulse stimulation), and then complete medium with 100 U/mL IL-2 was added (re-added at day 3 or day 4) and cultured for 7 days. Carboxyfluorescein succinimidyl ester (CFSE) labeling was done with CellTrace™ CFSE Cell Proliferation Kit (Thermo Fisher Scientific, Waltham, MA, USA): after resting, cells were labeled at a cell concentration of 15 × 10^6^ cells/mL and 1.5 μM CFSE for 5 min at room temperature and washed 3 times after which the stimulators were added and the cells were cultured for 5 days. For the determination of cytokine production in T cells, no IL-2 were added during culture and 2 μM monensin was added 4 h before harvest. For the CD107a assay: in the 4 h-stimulation assays, CD107a-PC7 (clone eBioH4A3, eBioscience) was added right after the addition of the stimulant, the cells were incubated for an hour after which monensin was added; in 1 day-stimulation assays, CD107a and monensin were added 4 h before harvesting the cells. For the assays with mevastatin, cells were pretreated with 2 μM mevastatin (Sigma-Aldrich) for 1 h, then proceeded to stimulation and CD107a staining. For BTN3A blocking tests, zoledronate-expanded (1 μM, 10–14 days, cryopreserved) PBMC were used to restimulate with the indicated stimulants. BTN3A (clone 103.2, kind gift from Imcheck Therapeutics; final concentration was 10 μg/mL) [20] and isotype control (IgG2A, 10 μg/mL) antibodies were added 1 h before stimulation. For apyrase (Sigma-Aldrich) treatment, each stimulant was incubated with 0.2 U/mL at 37 °C for 2 h, control stimulants were incubated at 37 °C for 2 h without apyrase. For heat treatment, mistletoe products were heated at 80 °C for 5 min.

### 2.3. Flow Cytometry and Antibody Reagents

Harvested cells were first washed with PBS, then labeled with 1000 times diluted Zombie NIR™ dye (Biolegend, San Diego, CA, USA) at room temperature for 20 min and washed with 0.1%BSA/PBS. For surface staining, cells were incubated with antibody mix at 4 °C for 15–20 min, then washed and resuspended with 1%PFA/PBS. For intracellular cytokine stainings, after surface staining, CytofixCytoperm™ (BD) was used to permeabilize cell membranes. Staining results were acquired either on CyAn ADP cytometer (Dako Cytomation) or LSRFortessa (BD); analysis was done using FlowJo software and R.

The following antibodies were used in this study: CD3-PB (clone UCHT1, BD), CD3-BV510 (UCHT1, BD), TCR γδ-APC (11F2, Miltenyi Biotec, Bergisch Gladbach, Germany), TCR Vγ9-PC5 (IMMU 360, Beckman Coulter, Brea, CA, USA), TCR Vδ2-FITC (IMMU 389, Beckman Coulter), CD4-V500 (RPA-T4, BD), CD4-BV510 (RPA-T4, BD), CD8-PC7 (SFCI21Thy2D3, Beckman Coulter), CD56-PE-CF594 (NCAM16.2, BD), CD69-PE (FN50, BD), IFNγ-V450 (B27, BD), TNFα-FITC (MAb11, BD), IL-17a-PE (eBio64DEC17, eBioscience). Dead cells were excluded (negative for Zombie NIR) and gated on CD3+ lymphocytes (Appendix A). Gating CD3+Vγ9+ lymphocytes identify the vast majority of Vγ9Vδ2 T cells in adult blood [21].

### 2.4. Cell Sorting, RNA Isolation, and CDR3 Analysis

PBMC were exposed to 10 nM HMBPP, 1 μM zoledronate, and 1000 μg/mL AbnobaViscum P for 1 day, washed 3 times, after which complete medium containing 100 U/mL IL-2 was added. IL-2 was added every 3-4 days during culturing for 14 days to allow an expansion of Vγ9Vδ2 T cells. Harvested cells were labeled with Zombie NIR™ dye (Biolegend) at room temperature for 10 min, and stained with CD3/TCR-γδ/TCR-Vγ9/TCR-Vδ2 antibodies at 4 °C for 15 min. Zombie NIR-CD3+γδ+Vγ9+ Vδ2+ T cells were sorted on FACS Aria III (BD) (purity range 94.4–100% (%Vγ9Vδ2 of T cells)). Cells were snap-frozen in liquid nitrogen and preserved at −80 °C.

RNA was isolated from sorted cells (~10000 cells each sample) with the RNeasy Micro Kit (Qiagen, Hilden, Germany) and CDR3γ and CDR3δ high-throughput sequencing was performed as described before [22]. Briefly, RNA was reverse transcribed via a template-switch cDNA reaction followed by a PCR amplifying the CDR3γ and CDR3δ regions. High-throughput sequencing of the generated amplicon products containing the TRG and TRD sequences was performed on an Illumina MiSeq platform using the V2 300 kit, with 150 base pairs (bp) at the 3′end (read 2) and 150 bp at the 5′end (read 1) [at the GIGA center, University of Liège, Belgium]. Raw sequencing reads from fastq files (read 1 and read 2) were past the quality check using fastqc (version 0.11.8, http://www.bioinformatics.babraham.ac.uk/projects/fastqc/). Then the sequences were aligned to reference V, D, and J genes from GenBank database specifically for ‘TRG’ or ‘TRD’ to build CDR3 sequences using the MiXCR software (version 3.0.3) [23]. Default parameters were used except to assemble TRDD gene segment where 3 instead of 5 consecutive nucleotides were applied as an assemble parameter. CDR3 sequences were then exported and analyzed using VDJtools software (version 1.2.1) using default settings [24]. Data for CDR3 length, treemaps and (D)J usage are generated by the VDJtools routine ‘annotate’; normalized Shannon Wiener Index by the routine ‘CalcDiversityStats’; Top shared clonotypes by the routine ‘TrackClonotypes’ and multidimensional scaling analyzing by the routine ‘ClusterSamples’) [24]. Sequences that were out of frame and contained stop codons were excluded from the analysis. Files generated from VDJtools were uploaded into Rstudio (version 1.1.463, R version 3.5.2) and analysis involving the following packages: Treemap (https://CRAN.R-project.org/package=treemap) to generate Treemap plots, ggplot2 [25] for data visualization and ggpubr (https://CRAN.R-project.org/package=ggpubr) for statistical analysis.

### 2.5. Data Availability

Fastq files of TRG and TRD sequences are deposited under SRA accession code PRJNA633174.

### 2.6. Statistics

Statistical analysis was performed with the R software package ggpubr (https://CRAN.R-project.org/package=ggpubr). Paired t-test was used for normally (determined by the Shapiro-Wilk test, *p* > 0.05) distributed data and with equal variances (determined by the Levene’s test, *p* > 0.05). Otherwise, Wilcoxon signed-rank test was used.

## 3. Results

### 3.1. AbnobaViscum but Not Iscador Mistletoe Extracts Induce Specific Expansion of Vγ9Vδ2 T Cells

We obtained four commercially available *Viscum album* L. (VA) extracts from two companies that derive them from the same host trees but use different preparatory methods: non-fermented extracts are AbnobaViscum Pini (AP) and AbnobaViscum Mali (AM) and fermented products are Iscador Malus (IM) and Iscador Pinus (IP). In order to perform short-term (4 h, 1 day) assays to assess T cell activation (CD69) and function (cytokine production and release of cytotoxic granules) and long-term (7 days) expansion cultures, we first titrated each extract to assess the potential cytotoxic effect and dose-response in PBMC cultures. After 1 day of culture, none of the extracts showed a cytotoxic effect (Appendix A, left panels). At 7 days however, all the VA extracts except IP showed a clear dose-dependent cytotoxicity (Appendix A, right panels). Heat treatment of mistletoe extracts [26] prevented the cell death induction in the long-term PBMC cultures (Appendix A). As heat-treatment is not performed on the mistletoe-extracts that are injected in cancer patients, we preferred to test an alternative method to prevent the cytotoxic effects in order to verify whether the heat-treatment was essential for possible effects on Vγ9Vδ2 T cells [26]. Exposure of the cells for one day to the VA extracts (at the highest concentration, 1000 µg/mL) followed by washing (‘pulse’), instead of continued exposure, prevented or reduced significantly the cytotoxic effects in longer term cultures (Appendix A). Thus, we included results from both heat treatment and pulse stimulation in the 7 days expansion cultures since both methods resulted in similar expansion levels (%Vγ9+ of CD3+ T cells, data not shown). Based on short-term activation experiments with the lymphocyte activation marker CD69, we selected 1000 μg/mL concentration of mistletoe extracts for further experiments (Appendix A). Both AbnobaViscum (AP and AM) and Iscador (IA and IM) VA extracts activated Vγ9+ γδ T cells, with no or minimal effects on Vγ9− γδ T cells and αβ T cells (Figure 1A). A different trend was observed for NK cells: here the activation was more pronounced with Iscador compared to AbnobaViscum extracts (Figure 1A). Surprisingly, while both AbnobaViscum and Iscador extracts activated Vγ9Vδ2 T cells (Figure 1A), and despite being derived from the same host trees, only AbnobaViscum extracts induced proliferation of Vγ9Vδ2 T cells (Figure 1B,C). AP was the strongest stimulant (Figure 1C; AP vs AM: *p* = 0.0078, Wilcoxon signed-rank test) and the expansion was highly restricted to Vγ9Vδ2 T cells (Figure 1B for CD3+ T cells, data not shown for CD3− NK cells). We therefore focused further on the stimulation of Vγ9Vδ2 upon treatment with AP. Of note, AP-induced expansion levels were similar to expansion levels observed with known Vγ9Vδ2 T cell stimulants (HMBPP, zoledronate) and individual AP-, HMBPP-, and zoledronate-induced expansions showed a strong correlation (Figure 1D).

In sum, although both AbnobaViscum (non-fermented) and Iscador (fermented) VA extracts can activate Vγ9Vδ2 T cells, only exposure to AbnobaViscum VA extracts results in their proliferation. This strong expansion was highly specific for Vγ9Vδ2 T cells.

### 3.2. AbnobaViscum Rapidly Stimulates the Release of Cytotoxic Granules and the Production of IFNγ and TNFα in Vγ9Vδ2 T Cells

As VA-extract drugs are used as complementary cancer therapy, we assessed the induction of the two main anti-cancer functions of Vγ9Vδ2 T cells: degranulation of their cytotoxic granules (by analyzing the surface expression of the granule-associated CD107a) and the induction of the cytokines IFNγ and TNFα. AP induced a rapid (4 h) and striking upregulation of CD107a, IFNγ and TNFα in Vγ9+ T cells, but not on Vγ9− T cells (Figure 2A–D). The release of cytotoxic granules and production of cytokines were largely co-expressed (Figure 2D). While some studies have ascribed an anti-tumor role for IL-17-producing γδ T cells like when they act in concert with immunogenic cell death-inducing chemotherapeutic drugs [27], in a range of other settings a pro-tumor role has been proposed [1,2,28]. Here, we could not find significant upregulation of this cytokine in Vγ9Vδ2 T cells upon exposure to AP (Appendix A). Of note, the stimulation kinetics of AP (4 h rather than 1 day) was similar to the stimulation kinetics of HMBPP (direct activation) but not with the kinetics of the indirect stimulatory compounds zoledronate and *sec*-butylamine (Figure 2E). In sum, AP rapidly stimulates the degranulation of cytotoxic granules and the production of the anti-cancer cytokines IFNγ and TNFα in Vγ9Vδ2 T cells but not within other T cells.

### 3.3. AbnobaViscum Stimulation of Vγ9Vδ2 T Cells is Direct and BTN3A-Dependent

Alkylamines such as *sec*-butylamine of edible plants and tea have been described as main Vγ9Vδ2 T cell-stimulating compounds [8,9] and to act, like aminobisphosphonates, indirectly by endogenous phosphoantigen (IPP) accumulation (10). Whether the same kind of ‘indirect’ Vγ9Vδ2 activating compounds or rather more direct mechanism are involved in the AP-induced activation is not clear. In order to verify the involvement of endogenous accumulation of IPP we used mevastatin, to inhibit HMG-Coenzyme A reductase activity upstream of IPP synthesis and thus to inhibit IPP production [29]. To our surprise, inhibiting IPP synthesis did not decrease the AP-induced stimulation (Figure 3A). As expected, direct HMBPP-induced activation was not influenced by mevastatin treatment as well, while *sec*-butylamine- and zoledronate-induced Vγ9Vδ2 T cell activation were inhibited (Figure 3A). These results rather indicate that AP contains (a) direct activating pyrophosphate compound(s). To address this further, pretreatment of AP with apyrase, that sequentially releases inorganic phosphate groups from phosphorylated molecules, completely abolished the Vγ9Vδ2 T cell response, while the same pretreatment of zoledronate and *sec*-butylamine did not influence their Vγ9Vδ2 T cell-activation potential (Figure 3B). To investigate further the mechanism of activation of Vγ9Vδ2 T cells by AP, we verified the involvement of BTN3A, that plays a crucial role in the phosphoantigen-mediated activation via the Vγ9Vδ2 TCR: the blocking BTN3A antibody 103.2 completely abolished AP-induced degranulation and cytokine production (Figure 3C). Thus overall, the AP-induced activation of Vγ9Vδ2 T cells does not depend on the accumulation of endogenous IPP production and is mediated via BTN3A.

### 3.4. The AbnobaViscum-Responsive Vγ9Vδ2 TCR Repertoire is Similar to the Phosphoantigen-Responsive Repertoire

Next, we wondered whether the BTN3A/Vγ9Vδ2 TCR-dependent AP stimulation induces a similar polyclonal Vγ9Vδ2 T cell response as HMBPP and zoledronate [30] or whether AP targets rather a subset of Vγ9Vδ2 T cells. In order to address this issue, we investigated the CDR3γ and CDR3δ repertoire by high-throughput sequencing of Vγ9Vδ2 T cells expanded with AP, and compared it to HMBPP- and zoledronate-expanded Vγ9Vδ2 T cells. The CDR3 length distributions were highly similar between AP-, HMBPP- and zoledronate-expanded Vγ9Vδ 2 T cells (Figure 4A). Compared to HMBPP- and zoledronate- expanded Vγ9Vδ2 T cells, AP-expanded CDR3γ and CDR3δ sequences showed the same diversity levels (Figure 4B), the same TRGJ and TRDJ usage (Figure 4C) and were highly shared (Figure 4D). This high sharing between the different treatments was mainly due to public clonotypes in the TRGV9 repertoire, i.e., shared between the different individuals (Figure 4D, left panels), while the shared AP/HMBPP/zoledronate TRDV2 repertoire was private for each individual (Figure 4D, right panels). This was further shown by a global analysis via multidimension scaling (MDS): the AP-, HMBPP- and zoledronate-expanded TRD repertoire were grouped in subject-based clusters (Figure 4E, right panel), highlighting again the similarities between the Vγ9Vδ2 TCR responses induced by AP, HMBPP and zoledronate.

In sum, the AP-responsive Vγ9Vδ2 TCR repertoire is similar to the direct (HMBPP) and indirect (zoledronate-induced IPP accumulation) phosphoantigen-responsive repertoire.

## 4. Discussion

Non-fermented mistletoe-extract drugs (AbnobaViscum) induced the specific expansion of Vγ9Vδ2 T cells, the rapid release of their cytotoxic granules and production of IFNγ and TNFα. All these features are known to be associated with anti-cancer activity [1,2]. AbnobaViscum has been shown to upregulate the expression of maturation markers on dendritic cells (DC), but failed to increase important cytokines such as IL-12p70 needed to stimulate and differentiate αβ T cells [31]. However, the promotion of IFNγ production by Vγ9Vδ2 T cells by AbnobaViscum may indirectly promote the full maturation of DC including IL-12p70 production [32,33,34,35] that is initiated by the direct DC-stimulation by mistletoe-derived lectins [31]. These fully mature DC could then in turn promote the development of (tumor) antigen-specific αβ T cell responses [35]. IL-17 production by γδ T cells has been associated with the promotion of tumor development [2,28], but we did not find evidence for significant production of this cytokine by Vγ9Vδ2 T cells upon AbnobaViscum exposure.

The Vγ9Vδ2 T cell response appeared to be specific to the type of preparation of the mistletoe extracts. Despite being derived from the same host trees (pine or apple tree), the bacterial-fermented extracts from Iscador did not result in the expansion of Vγ9Vδ2 T cells. This was rather unexpected as the fermentation process could be a source of bacterial phosphoantigens [5,36] and thus indicates that these bacteria are not a source of Vγ9Vδ2 T cell-activating phosphoantigens in mistletoe-extract drugs. Possibly, the fermentation process leads to a degradation of mistletoe-derived pyrophosphates and rather induce compounds that are stimulatory for NK cells. Indeed, it has been suggested that Iscador preparations are stimulatory while AbnobaViscum preparations are inhibitory for NK cells [15,18,37]. This is in line with our observation that Iscador induced higher CD69 expression on NK cells than AbnobaViscum. It is known that fermented and non-fermented mistletoe extracts can be very different in terms of their composition [38] and our recent metabolomics analysis on a series of mistletoe-extract drugs indicate that the composition of the extracts is much more dependent on the producer (company) than on the host tree (manuscript in preparation). Thus, the main immune cell target for fermented mistletoe extracts such as Iscador could be NK cells. The CD69 induction observed on Vγ9Vδ2 T cells upon exposure towards Iscador extracts could therefore be secondary to NK cell activation and thus rather a bystander effect. While this bystander effect could be sufficient for the increased cell surface expression of the sensitive activation marker CD69, this effect may not be sufficient for the more robust signaling needed to induce expansion of the Vγ9Vδ2 T cells [39] and thus provide a possible explanation why we did not observe any expansion of Vγ9Vδ2 T cells upon exposure to Iscador extracts.

Alkylamines that are present in plants such as *sec*-butylamine have been shown to activate Vγ9Vδ2 T cells indirectly by inhibiting the enzyme farnesyl pyrophosphate synthase resulting in the upregulation of endogenous phosphoantigens [8,10]. However, using approaches that verified the role of endogenous and exogenous phosphoantigens in the activation of Vγ9Vδ2 T cells by AbnobaViscum, we show here that the activation is mediated directly and thus not depend on the intracellular accumulation of phosphoantigens. Our findings are in line with the observed sensitivity towards alkaline phosphatase of heat-treated mistletoe extract-induced γδ T cell expansion [26]. The absence of expansion without heat treatment [26] is likely due to the cytotoxic effects of the mistletoe extracts in longer term cell cultures needed to study proliferation. Indeed, we showed that the mistletoe-extract drugs as such (thus without heat treatment) are sufficient to stimulate Vγ9Vδ2 T cells. In general, Vγ9Vδ2 T cell-activating phosphoantigens can be derived from the mevalonate pathway or the non-mevalonate pathway, the latter also known as methylerythritol phosphate (MEP) pathway. Most organisms only use one of the two pathways for their isoprenoid synthesis. The MEP pathway is the one present in most (pathogenic) eubacteria and parasites of the phylum Apicomplexa, but it is absent from archaebacteria, fungi, and animals, which synthesize their isoprenoids exclusively through the operation of the mevalonate pathway. By contrast, plants use both the MEP pathway and the mevalonate pathway for isoprenoid biosynthesis, although they are localized in different compartments: the MEP pathway is active in the plastids while the mevalonate pathway in the cytosol [40]. These plastids are likely derived from once free-living bacteria by endosymbiosis [41,42]. *Viscum album* L, used for the generation of mistletoe-extract drugs, contains the gene expression profile of the enzymes needed for the MEP pathway [43]. HMBPP, a MEP pathway-derived Vγ9Vδ2 T cell-activator, is up to 10,000 times more potent than the mevalonate pathway-derived Vγ9Vδ2 T cell-activator IPP and is thus described as a main compound allowing Vγ9Vδ2 T cells to sense cells infected with bacteria or parasites such as *Plasmodium* [5,44,45]. We propose that AbnobaViscum contains HMBPP or metabolites with a similar structure derived from the plastid-derived MEP pathway and thus that its administration mimics the presence of bacterial- or parasite-derived HMBPP resulting in the stimulation of Vγ9Vδ2 T cells that are cross-reactive with cancer cells [46]. Since the AbnobaViscum-induced Vγ9Vδ2 T cell stimulation was completely dependent on BTN3A, we propose that mistletoe-derived phosphoantigens such as HMBPP act in a direct and rapid manner via this ubiquitously expressed butyrophilin [6,47]. These phosphoantigens could act in concert with other mistletoe-derived compounds such as lectins (glycoproteins) and viscotoxins (polypeptides) targeting other immune cells such as DC and NK cells [31,48].

It has been recently described that both HMBPP and zoledronate stimulate polyclonal TCR responses as assessed by high-throughput sequencing of the Vγ9Vδ2 TCR repertoire [30]. We wondered whether AbonabaViscum stimulated similar TCR responses as HMBPP and/or zoledronate, or whether it would act only on a restricted Vγ9Vδ2 TCR repertoire as described for the tuberculosis vaccine BCG [49]. The AbnobaViscum-expanded Vγ9Vδ2 TCR repertoire showed a high level of similarity with the HMBPP-and zoledronate-expanded repertoire at the level of CDR3 length, CDR3 diversity, and (D)J gene segment usage. Furthermore, the same top expanded TRGV9-associated CDR3 sequences could be found among AbnobaViscum-, HMBPP- and zoledronate- expanded TCR repertoires, that were highly shared among subjects (i.e., public). In contrast, the TRDV2-response was highly private: the expanded TCR repertoire was specific for each subject, confirming previous studies [30,50,51], but again the same top TRDV2-associated CDR3 sequences could be found in the AbnobaViscum-, HMBPP- and zoledronate-expanded TCR repertoires. Thus, AbnobaViscum appears to act on the same polyclonal Vγ9Vδ2 TCR repertoire in adult peripheral blood as does HMBPP and IPP (upregulated by zoledronate), and not on a small subset of Vγ9Vδ2 T cells. Despite significant recent progress regarding the molecular basis of phosphoantigen recognition by Vγ9Vδ2 T cells, it is still not known to which molecular structures the CDR3 regions of both the TRGV9 and TRDV2 chains bind [52,53]. It remains thus unclear why the TRGV9 and TRDV2 repertoire of adult Vγ9Vδ2 T cells, also after expansion with AbnobaViscum, are public and private respectively.

While bone targeting of aminobisphosphonates can be useful in the treatment of bone-related diseases, including cancer metastasis to the bones, this is rather a disadvantage for the treatment of most cancers that do not show bone metastasis. Furthermore, treatment with bisphosphonates such as zoledronate can lead to medication-related osteonecrosis of the jaw, a serious adverse reaction [54]. Thus, alternatives are being developed such as the ex-vivo expansion of Vγ9Vδ2 T cells in order to re-infuse in cancer patients or the development of Vγ9Vδ2 T cell-activators with potential improved pharmacokinetic properties [1,55,56]. However, these compounds are till now not administrated in vivo and the injection of expanded Vγ9Vδ2 T cells have only been performed in the context of small phase 1/2 clinical trials [1]. In contrast, the administration of mistletoe-extract drugs is safe and about 500,000 cancer patients each year in Germany receive this treatment [14]. Our in vitro data regarding the stimulation of anti-cancer Vγ9Vδ2 T cells by AbnobaViscum encourages the inclusion of these cells in future immunophenotyping studies upon AbnobaViscum treatment in vivo. Such immunophenotyping data and their possible correlation with clinical outcome will allow the stratification of cancer patients and are expected to provide insight into the controversial anti-cancer activities of mistletoe-extract drugs in cancer patients.

## Figures and Tables

**Figure 1 cells-09-01560-f001:**
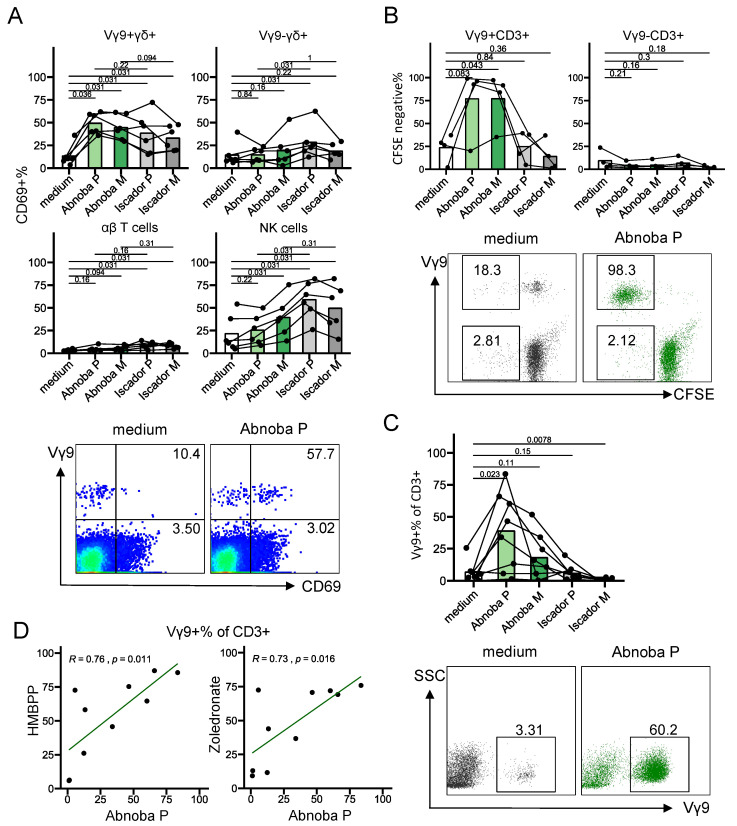
AbnobaViscum but not Iscador mistletoe extracts induce specific expansion of Vγ9Vδ2 T cells. (**A**) Percentage of CD69 expression on different cell types after stimulation with different mistletoe extracts for 1 day. Upper left: Vγ9+ γδ T cells (CD3+γδ+Vγ9+); upper right: Vγ9− γδ T cells (CD3+γδ+Vγ9−); lower left: αβ T cells (CD3+γδ−); lower right: natural killer (NK) cells (CD3−CD56+). Lines connect the same subjects (*n* = 6), bars indicate mean values. Values on the graphs indicate *p* values (obtained with the Wilcoxon signed-rank test). Bottom panels show representative flow cytometry plots (gated on CD3+ T cells), numbers indicate percentages of CD69+ cells, expressed as a percentage of Vγ9+CD3+ cells (top) and as a percentage of Vγ9−CD3+ cells (bottom). (**B**) Percentage of CFSE-negative Vγ9+ T cells (CD3+Vγ9+, upper left) and Vγ9− T cells (CD3+Vγ9−, upper right) after stimulation for 5 days with different mistletoe extracts. Lines connect the same subjects (*n* = 4), bars indicate mean values. Values on the graphs indicate *p* values (obtained with the paired *T*-test). Bottom panels show representative flow cytometry plots (gated on CD3+ T cells), numbers indicate percentages of CFSE-negative cells, expressed as a percentage of Vγ9+CD3+ (top) and as a percentage of Vγ9−CD3+ (bottom). (**C**) Percentage of Vγ9+ cells (of total CD3+ T cells) after stimulated with different mistletoe extracts for 7 days. Lines connect the same subjects (*n* = 8), bars indicate mean values. Values on the graphs indicate *p* values (obtained with the Wilcoxon signed-rank test). Bottom panels show representative flow cytometry plots (gated on CD3+ T cells), numbers indicate percentages of positive cells in the indicated gates. (**D**) Correlation between AbnobaViscum P- and HMBPP-, and zoledronate-induced expansion (7 days). Each dot represents one subject (*n* = 10).

**Figure 2 cells-09-01560-f002:**
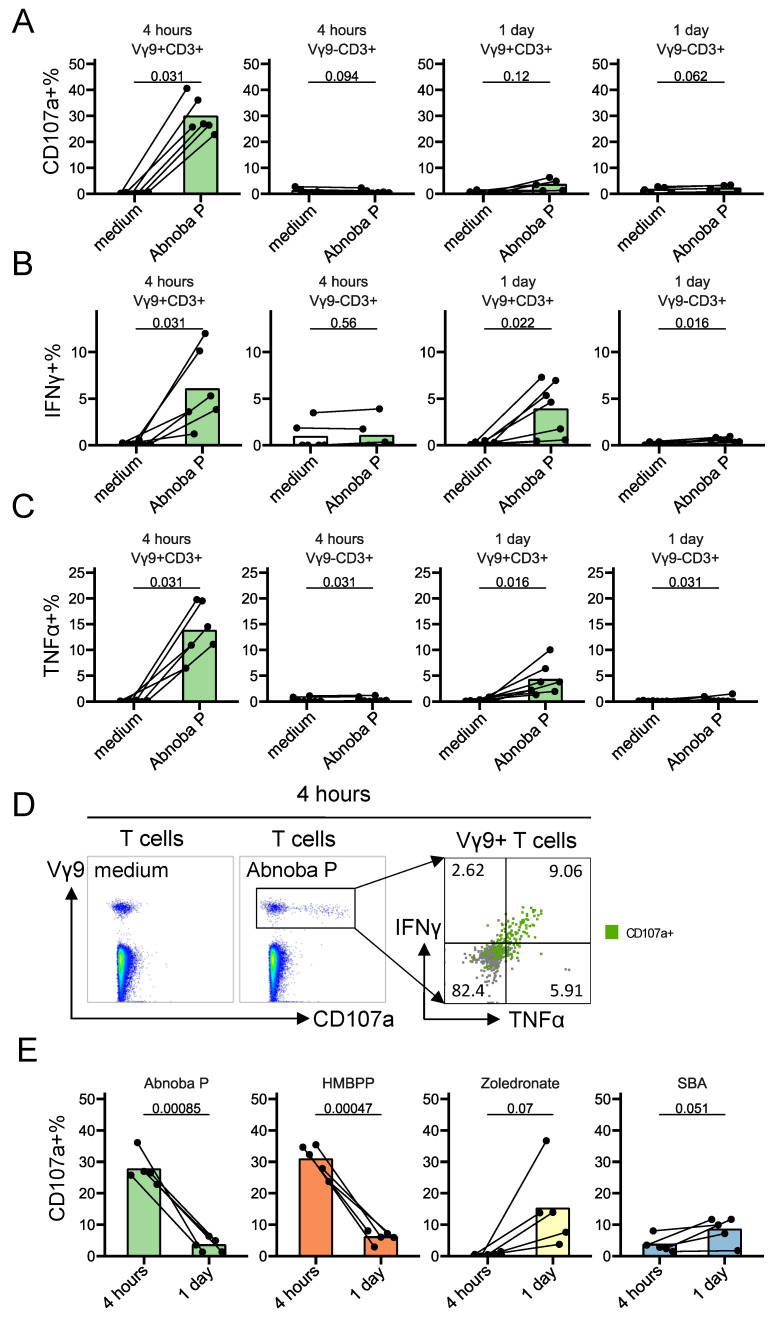
AbnobaViscum rapidly stimulate the release of cytotoxic granules and the production of IFNγ and TNFα in Vγ9Vδ2 T cells. (**A**–**C**) CD107a (A), IFN-γ (B) and TNF-α (C) expression on Vγ9+ T cells and Vγ9− T cells after AbnobaViscum Pini (Abnoba P) stimulation. Lines connect the same subjects (*n* = 6), bars indicate mean value. Values on the graphs indicate *p* values (obtained with the Wilcoxon signed-rank test). (**D**) Representative flow cytometry plots (4 h stimulation): the first two plots are gated on T cells (medium control on the left, Abnoba P on the right), the third plot is gated on Vγ9+ T cells (Abnoba P), illustrating CD107a, IFNγ and TNFα co-expression (**E**) Kinetics of CD107a expression on Vγ9+ T cells by Abnoba P, HMBPP, zoledronate and sec-butylamine (SBA). Lines connect the same subjects (*n* = 5), bars indicate mean values. Values on the graphs indicate *p* values (obtained with the paired *T*-test).

**Figure 3 cells-09-01560-f003:**
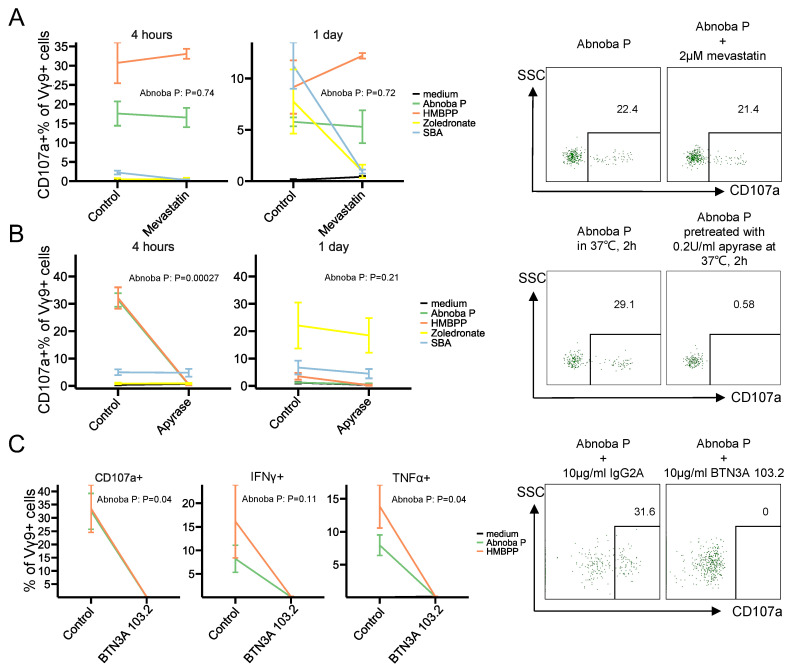
AbnobaViscum stimulation of Vγ9Vδ2 T cells is direct and BTN3A-dependent. (**A**) CD107a expression on Vγ9+ T cells upon mevastatin treatment within each stimulation. Lines connect the mean values between control and mevastatin treatment within the same stimulation, error bars show mean±sem (*n* = 3). Representative flow cytometry plots after 4 h stimulation are on the right of the graphs (gate on CD3+Vγ9+ T cells). (**B**) CD107a expression on Vγ9+ T cells upon apyrase treatment within each stimulation. Lines connect the mean values between control and apyrase treatment within the same stimulation, error bars show mean±sem (*n* = 5 for 4 h, *n* = 3 for 1 day). Representative flow cytometry plots after 4 h stimulation are on the right of the graphs (gate on CD3+Vγ9+ T cells). (**C**) CD107a (left), IFNγ (middle), TNFα (right) expression in Vγ9+ T cells upon blocking BTN3A within each stimulation for 4 h. Lines connect the mean values between isotype control and BTN3A 103.2 mAb within the same stimulation, error bars show mean±sem (*n* = 3). Representative CD107a stainings (4 h stimulation) after are on the right of the graphs (gate on CD3+Vγ9+ T cells). Values on the graphs indicate *p* values (obtained with paired *T*-test).

**Figure 4 cells-09-01560-f004:**
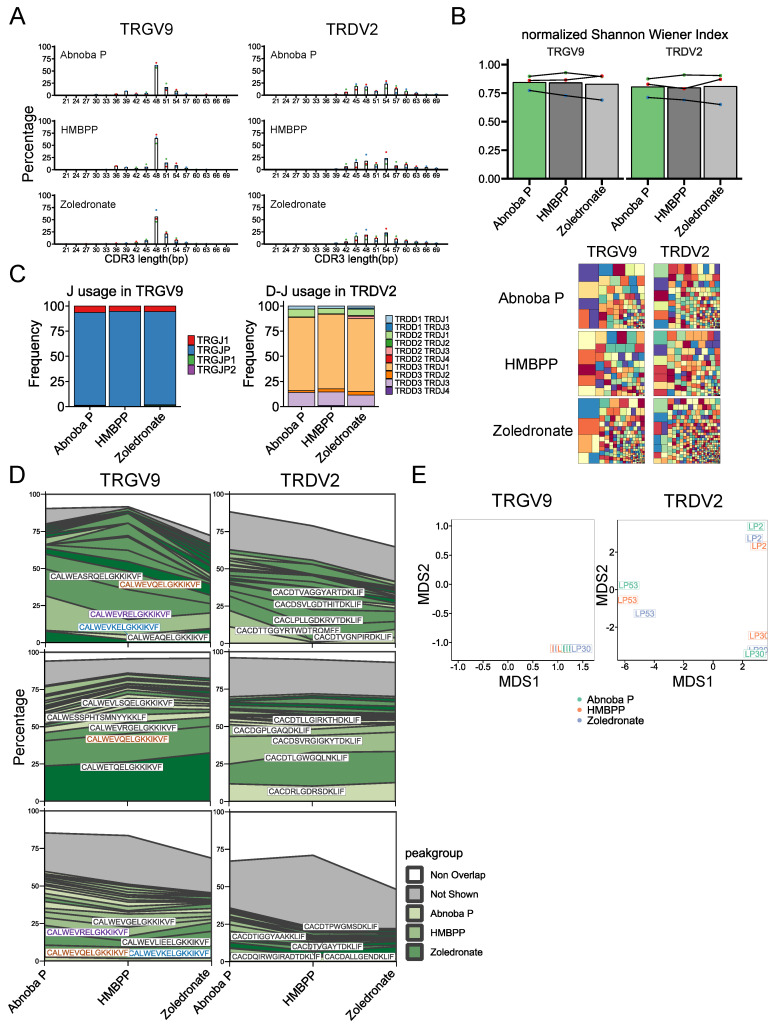
The AbnobaViscum-responsive Vγ9Vδ2 TCR repertoire is similar to the phosphoantigen-responsive repertoire. (**A**) Distribution of CDR3 length for TRGV9- and TRDV2-containing CDR3 sequences after expansion with the indicated Vγ9Vδ2 T cell stimulators. Each color of the dots represents the same subject, bar indicates mean percentage for each CDR3 length (expressed in nucleotides). (**B**) Diversity of TRGV9- and TRDV2- containing CDR3. Normalized Shannon Wiener index: each color of the dots represents the same subject, lines connect each subject, bars indicate mean value. Representative treemaps for the indicated stimulators are below the graphs: each small square represents a CDR3 sequence of which the size is related to the frequency of the sequence within the repertoire within each sample (rectangle colors are chosen randomly and do not match between plots). (**C**) Mean J gene segment usage in TRGV9-containing CDR3 (left) and mean D-J gene segment usage in TRDV2-containing CDR3 sequences (right) (*n* = 3). (**D**) Sequence overlap between AP-, HMBPP- and zoledronate-induced expansions for TRGV9-containing CDR3 (left) and TRDV2-containing CDR3 (right). The top 20 sequences are filled with different green shades, the remaining overlapping sequences are indicated in grey and the non-overlapping sequences are in white. Top 5 shared sequences are provided on the plots for each subject: colored sequences occur in more than one subject while black sequences indicate unique sequences. (**E**) Multidimensional scaling analysis of TRGV9-containing CDR3 sequences (left) and TRDV2-containing CDR3 sequences (right). Colors indicate each expansion; the subject number is indicated within each small square.

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
