# Peer review of "Mistletoe-Extract Drugs Stimulate Anti-Cancer Vγ9Vδ2 T Cells"

_cells, 2020, doi:10.3390/cells9061560_

Round 1
Reviewer 1 Report
This manuscript describes a puzzling observation where the use of European mistletoe (an hemi-parasite plant) was able to stimulate the innate like Vgamma 9Vdelta 2 T cells. These effects were dependent both on the statins (meaning a role of IPP) and BTN (via the use of the BTN3A blocking mAb). The effects led to the specific expansion and CD107 degranulation and cytokine production by these cells.
Surprinsingly the TcR elicited was moreless similar to the one innduced by phosphoantigens.
These highly puzzling data are clearly written and convincing.
It would be interesting to follow what might be the next steps of the experiments
Author Response
We thank the reviewer for their positive comments.
Reviewer 2 Report
Vγ9Vδ2 TCR (Vg9dT) cells possess potent anti-tumor activity making them attractive targets of cancer immunotherapeutic approaches. In this manuscript, “Mistletoe-Extract Drugs Stimulate Anti-Cancer Vγ9Vδ2 T Cells”, Ma et. al have investigated the in vitro response of human PBMC derived T cells towards mistletoe-extract (ME), and ME derived drugs which are used as complementary cancer therapy in European countries. Towards this end, the authors have used a combination of in vitro functional assays, flow cytometry and high-throughput TCR sequencing to elucidate the main target cells of ME, and mechanism of action of ME which contributes to the observed anti-cancer activity of these extracts.
The authors demonstrate that both fermented and non-fermented ME lead to the activation of immune cell types in the peripheral blood including Vg9dT cells and NK cells. However, only the non-fermented ME induces expansion, specifically of the Vg9dT cells. They also show that the non-fermented ME induces potent effector function in Vg9dT cells. This polyfunctionality of Vg9dT cells observed following stimulation with ME was similar to ME-derived drugs. Indeed, they show that both ME and HMBPP induced activation of Vg9dT cells does not depend on the accumulation of endogenous IPP production and is mediated via BTN3A. Lastly, the authors also show a high degree of overlap and diversity of TCR repertoire of ME expanded Vg9dT cells with TCR repertoire ME derived drug expanded Vg9dT cells further demonstrating the similarity and potency of ME and ME derived drugs.
Identification of the specific targets of different ME products (non-fermented vs. fermented) and the mechanism of action of these extracts is of great interest to the field as if not only provides insight into the anti-cancer activities of ME derived drugs, it also argues a case for focusing on Vg9dT cells responses in patients receiving ME derived drugs for cancer therapy. This can potentially inform the parameters included in clinical trials and can also lead to the development of better drugs in future.
The manuscript has only a few minor issues:
- Line 60-61, it is not entirely clear what the authors mean by, “These and other γδ T cell-reactivities towards plant-derived compounds are thought to play an important role in the prevention of cancer development”. Please clarify.
- The rationale for only focusing on gdT cells in this study is not very clear. NK cells also demonstrate appreciable activation following exposure to ME, albeit much differently. Please add some explanation of why Vγ9Vδ2 T cells were selected for the remainder of the study.
- Line 228-229, while figure 2D shows that activated gdT cells express CD107a, IFNG and TNFa, it doesn’t really convey “co-expression”. It would be better to show bi-plots of CD107a gated gdT cells with IFNg and TNFa on x- and y-axis to demonstrate co-expression of these effector molecules. Maybe also include a “polyfunctionality” plot to show the highly potent effector-like state of these activated cells.
- Figure 2E, the legend is not required.
- Please add statistics to the analysis presented in Figure 3.
- In figure 4, description of how data was generated should be omitted from the figure legend and moved to the materials and methods section.
- Figure 4B, legend for treatment type not required. Instead, please indicate that the dots are donors 1, 2 and 3.
- Line 352, can the authors please provide a reference for “more robust signaling needed to induce expansion of the Vγ9Vδ2 T cells”.
- Need to include data from additional donors (at least 3, ideally 5) for experiments shown in supplementary figure 3.
- The TCR sequencing data is very exciting. If possible, can the authors comment more on the implications of high diversity of TCR clonotypes, the significant overlap of TRGV9-CDR3 repertoire observed between the treatment groups, and why do they think that the TRDV2-response was highly private in the discussion section.
